# Separate and Reconstruct:
# Asymmetric Encoder-Decoder for Speech Separation

**Ui-Hyeop Shin**     **Sangyoun Lee**     **Taehan Kim**     **Hyung-Min Park**

Department of Electronic Engineering, Sogang University, Seoul, Republic of Korea
`{dmlguq123,leesy0882,taehank,hpark}@sogang.ac.kr`

https://dmlguq456.github.io/SepReformer_Demo

## Abstract

In speech separation, time-domain approaches have successfully replaced the time-frequency domain with latent sequence feature from a learnable encoder. Conventionally, the feature is separated into speaker-specific ones at the final stage of the network. Instead, we propose a more intuitive strategy that separates features earlier by expanding the feature sequence to the number of speakers as an extra dimension. To achieve this, an asymmetric strategy is presented in which the encoder and decoder are partitioned to perform distinct processing in separation tasks. The encoder analyzes features, and the output of the encoder is split into the number of speakers to be separated. The separated sequences are then reconstructed by the weight-shared decoder, which also performs cross-speaker processing. Without relying on speaker information, the weight-shared network in the decoder directly learns to discriminate features using a separation objective. In addition, to improve performance, traditional methods have extended the sequence length, leading to the adoption of dual-path models, which handle the much longer sequence effectively by segmenting it into chunks. To address this, we introduce global and local Transformer blocks that can directly handle long sequences more efficiently without chunking and dual-path processing. The experimental results demonstrated that this asymmetric structure is effective and that the combination of proposed global and local Transformer can sufficiently replace the role of inter- and intra-chunk processing in dual-path structure. Finally, the presented model combining both of these achieved state-of-the-art performance with much less computation in various benchmark datasets.

## 1   Introduction

For the well-known cocktail party problem [14, 3], single channel speech separation [30] has been improved since the introduction of time-domain audio separation network (TasNet) [46, 47], which processes the audio separation in the latent space instead of the short-time Fourier transform (STFT) domain, as shown in Figure 1(a). In particular, in most speech separation methods, the process of separating the feature sequence for each speaker is typically positioned at the final stage of the network, as shown in Figure 1(b), which we refer to as *late split*. Therefore, a single feature sequence must encode all the information for all speakers to be separated. In addition, experimental results have shown that TasNet employing a convolution-based audio encoder/decoder performs better when the kernel length of the audio encoder is shortened [47, 45], which requires modeling a long sequence. Indeed, expanding the sequence in channel and temporal dimensions is necessarily beneficial, since the separation process must include all the information for all speakers in the feature sequence.

As a solution to modeling long sequences, DPRNN [45] was proposed using a dual path model, in which it segments long sequences into chunks and processes in terms of intra-chunk and inter-chunk

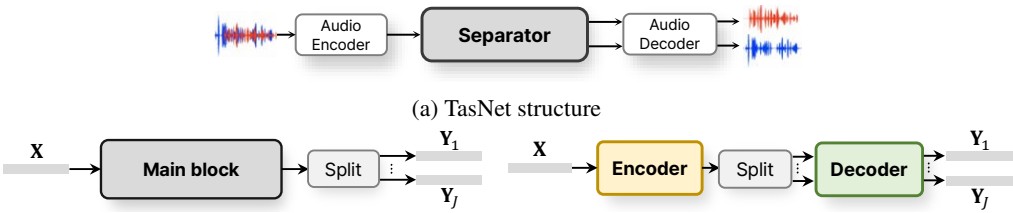

(a) TasNet structure

(b) Conventional separator design using a late split

(c) Proposed separator design using an early split

Figure 1: **Block diagrams of (a) TasNet and separator designs of the (b) conventional and (c) proposed networks.** The proposed network consists of separation encoder and reconstruction decoder based on weight sharing. After an encoder, separated features are independently processed by a decoder network.

to model the local and global contexts. As a result, due to promising performances in modeling long sequences, many TasNet-based approaches have adopted the dual-path model and repeatedly achieved state-of-the-art performances in monaural speech separation [9, 66, 37, 38, 92, 57, 60, 51]. Meanwhile, some studies have tackled the high computational complexity of long sequences in the time domain approach and proposed using multi-scaled sequence models based on the recurrent or stacked U-Net structure [70, 32, 42]. They reduced the computations to some extent, however, they still could not show competitive performance compared to the dual-path method.

However, most studies have focused on handling long sequences rather than addressing the fundamental inefficiency of TasNet's late split structure, where a single feature sequence must encode all speaker information without discrimination, creating an information bottleneck. Also, forcing the separator to generate all separated features at once before the audio decoder makes the task difficult and can easily lead to local minima during training. To alleviate this challenge, we propose a more intuitive approach: expanding the feature sequence to include a dimension for the number of speakers in advance, as an *early split*. By splitting features earlier in the separator (Figure 1(c)), we adopt an asymmetric strategy where the encoder and decoder perform distinct roles. The encoder process a single feature sequence before the split layer, similar to conventional separators. After splitting into multiple sequences, the decoder focuses on capturing *discriminative* characteristics between features using weight-sharing blocks [4, 11]. By employing this early split with a shared decoder (ESSD) structure, we ease the burden on the separator's encoder. This approach aligns with common practices in multi-channel speech processing, where processing is divided into two stages. Coarse separation is achieved through spatial filtering [31, 85, 24], followed by post-enhancement to refine results [80, 8, 81, 41].

Furthermore, dual-path model itself also has redundancy because the segmentation process may increase the amount of computation by twice when the overlap between adjacent chunks is set to 50%. Also, the inter-chunk blocks in the dual-path model are inefficient because their role is mainly to capture the global context. Therefore, we design unit blocks for both global and local processing, integrating them effectively to replace the dual-path model and directly process long sequences without chunking. Both of global and local blocks are based on Transformer block structure [23] where multi-head self-attention (MHSA) module and feed-foward network (FFN) module are stacked. As a global Transformer block, we modified MHSA module in Transformer block as an efficient gated attention (EGA) module to mainly capture the global dependency without redundancy. On the other hand, as a local block, the MHSA is replaced with convolutional local attention (CLA) to capture local contexts.

Consequently, we present the *Separation-Reconstruction Transformer (SepReformer)* for more efficient time-domain separation. Based on the ESSD framework and efficient global and local Transformer unit block, the SepReformer employs an asymmetric encoder-decoder structure with skip connections based on a temporally multi-scaled sequence model. The encoder processes a single feature at different temporal resolutions, and each intermediate feature is used for skip connection. The decoder then gradually reconstructs fine-grained information from the temporal bottleneck features, focusing on the discriminative characteristics of separated speech with auxiliary loss. To achieve this, the weight-shared decoder is trained to discriminate between the features separated by the encoder. In addition, a cross-speaker block is utilized in the decoder to facilitate information interaction between sequences, as described in [15, 40]. Furthermore, we design unit blocks for both global and local processing, integrating them effectively to replace the dual-path model and directly process long sequences without chunking. The experimental results demonstrated that the ESSD is

effective especially with a small network. Also, comprising the separator network with the proposed global and local blocks can sufficiently replace the inter- and intra-chunk processing in dual-path structure, suggesting effectiveness for long feature sequence processing in speech separation. As a consequence, the proposed SepReformer that includes both of these achieved state-of-the-art (SOTA) performance with much less computation than before in various benchmark datasets.

## 2 Related Works

**TasNet** Conventional source separation has been performed in the STFT domain [30, 36, 12, 44]. In the time-frequency representation, a separator is modeled to estimate mask values or direct output representations. Then, the inverse STFT (iSTFT) is operated for the output representations to obtain separated signals [30, 36]. On the other hand, TasNet [46] replaces STFT with a 1D convolutional layer. Based on the encoder representation, mask values or direct output representations [65, 59, 40] are obtained in the separator. Then, the output representations are decoded by the audio decoder of 1D transposed convolution instead of iSTFT. Also, unlike the STFT, the convolutional encoder turns out to work well in a much shorter kernel size. Therefore, TasNet requires the separator to process the much longer sequences. Therefore, instead of an LSTM-based separator [46], Conv-TasNet [47] is proposed based on a temporal convolutional network (TCN) [71, 39] to design a separator for longer sequence, showing impressive separation results.

**Dual-path model for long sequence** After Conv-TasNet, the dual-path model is extensively employed to handle long sequences. In the dual-path model, the sequence is segmented into smaller chunks, and the sequence is processed alternately as intra-chunk and inter-chunk, effectively interleaving between local and global contexts. This dual-path strategy has shown promising performance in TasNet and has been repeatedly adopted [9, 66, 37, 38, 92, 57, 60, 51, 53, 34]. Especially, it is shown that, compared to various efficient attention mechanisms [76, 2, 35], using the dual-path model with the original self-attention mechanism of Transformer [72] is effective for long sequence [67] in speech separation. However, modeling with the dual-path method can double computational costs due to the 50% overlap between adjacent chunks. The inter-chunk blocks in this model are somewhat redundant since they mainly capture global context. To reduce this redundancy, the quasi-dual-path network (QDPN) [59] replaces inter-chunk processing with downsampling. Inspired by QDPN, we design EGA and CLA modules to capture the global and local contexts without chunking process.

**Multi-scale model for efficiency** Instead of the dual-path model, based on U-Net structure [61], some studies have suggested using multi-scaled sequence model [65, 49, 21, 32, 42, 7]. SuDoRM-RF model [70] used a stacked U-Net structure to reduce the computational cost. The SuDoRM-RF approach can be regarded as a substitution of the TCN block in Conv-TasNet with U-ConvBlock as U-Net sub-block. Although SuDoRM-RF reduces the computational cost, it still has the disadvantages of having a fixed receptive field size and not considering the global context. More recently, TDANet [42] has efficiently improved performance with top-down attention and unfolding as in A-FCRNN [32]. However, these conventional methods with multi-scaled sequences prefer stacked or recurrent structures with U-Net sub-block to improve performance. Instead, we consider a single U-Net architecture and explicitly divide the roles of encoder and decoder as separation and reconstruction.

**Discriminative learning** Weight-sharing neural networks widely used in modern contrastive learning [11, 6, 25] including speaker verification [90, 58, 17]. On the other hand, some studies on speech separation proposed to exploit speaker identity as discriminative information to address the case that the similar voices are mixed [52, 51, 89]. Therefore, we utilized the weight-shared network to reconstruct separated speech by extracting distinct speech representations for corresponding speakers. To separate the mixture, weight-shared decoder directly learns to focus discriminative features without the need for additionally designed, for example, speaker loss using an additional speaker embedding extractor. As a result, based on discriminative learning, the weight-shared network in the decoder strengthens the dominant speaker's components on each separated sequence, respectively.

## 3 Method

### 3.1 Overall pipeline

When input mixture $\mathbf{x} \in \mathbb{R}^{1 \times N}$, the 1D convolution audio encoder, followed by GELU activation [28], encodes $\mathbf{x}$ to the input representation, as $\mathbf{X} = \mathcal{E}(\mathbf{x}) \in \mathbb{R}^{F_o \times T}$, where $F_o$ and $T$ denote the number of convolutional filter of encoder and the number of frames, respectively. The kernel and stride size are

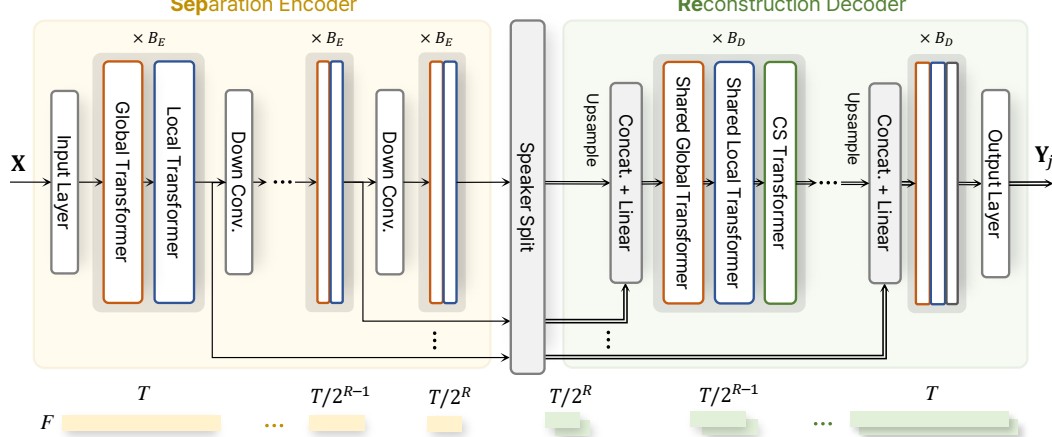

Figure 2: **The architecture of the separator in the proposed SepReformer.** The separator consists of three parts: separation encoder, speaker split module, and reconstruction decoder.

$L$ and $H$, respectively. Then, the $J$ output representations $\mathbf{Y}_j$ are estimated from the separator and decoded by the audio decoder, expressed as $\hat{s}_j = \mathcal{D}(\mathbf{Y}_j) \in \mathbb{R}^{1 \times N}, 1 \leq j \leq J$. Following the recent works [60, 79], we design the separator to directly map the output signals instead of masking.

## 3.2 Architecture of separator

The detailed architecture of the separator of the proposed SepReformer is illustrated in Figure 2. The separator is constructed on the basis of the ESSD framework with a separation encoder and a reconstruction decoder in temporally multi-scaled U-Net structure.

**Separation encoder** The input representation is first projected to $F$ dimension by the input layer. The input layer is composed of the linear layer and Layer Normalization (LN) [1] applied to each frame independently. In the encoder, the projected feature sequence is successively downsampled $R$ times from the sequence length of $T$ to $T/2^R$. The downsampling is performed by a 1D depth-wise convolution (Dconv) layer with a stride of 2 and a kernel size of 5, followed by Batch Normalization (BN) and GELU activation [28]. Each encoder stage processes the single sequence feature by $B_E$ stacks of global and local Transformer blocks.

**Speaker split** The encoded features in all stages of the encoder are expanded by the number of speakers $J$ to transmit the speaker-wise features from the encoder to the decoder. Therefore, the speaker split layer is placed in the middle, and it commonly separates the intermediate encoder features used for skip connections as well as the bottleneck feature. As shown in Figure 3, this module consists of two linear layers with gated linear unit (GLU) activation [20]. Each feature is then normalized by LN and processed by the decoder.

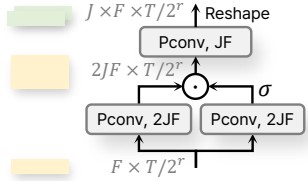

Figure 3: Speaker split module

**Reconstruction decoder** For temporal reconstruction, the upsampled sequence feature from the previous stage is concatenated with the skip connection followed by linear layer. Then, $B_D$ stacks of global and local Transformer blocks process the $J$ feature sequences as a weight-sharing network to discriminate between the separated features. By incorporating the separation objective function into the weight-sharing decoder, the network directly learns to capture the discriminative features. Then, the output of the last decoder stage is projected back to $F_o$ dimension by an output layer. The output layer consists of two linear layers with GLU activation.

**Cross-speaker (CS) Transformer** During the discrimination process by the weight-sharing decoder, speech elements can be mistakenly clustered into other speaker channels. As a result, it would be beneficial to attend to each other in order to effectively recover the distorted speech elements. Therefore, to improve the interaction of contexts between speakers within the decoder, we incorporate a Transformer-based CS module as in [15, 40]. Based on MHSA module without positional encoding, the CS block performs an attention operation on speaker dimension while temporal dimension is processed independently. Therefore, the CS block learns to identify the interfering components of the opposing sequences within the same temporal frame. For convenience, we call ESSD with CS as a separation-and-reconstruction (SepRe) method.

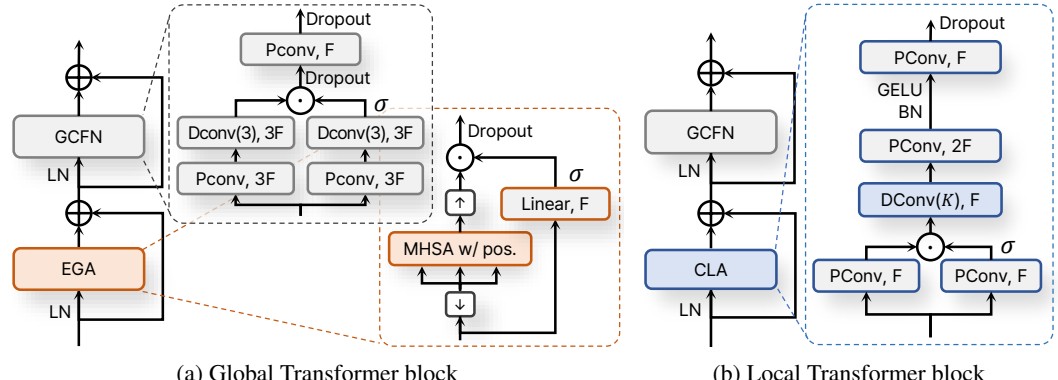

Figure 4: **Block diagrams of global and local Transformer for sequence processing**. ↓ and ↑ in EGA denote downsampling with average pooling and upsampling with nearest interpolation. Note that the point-wise convolution (Pconv) layer performs an equivalent operation to the linear layer as channel mixing. The hidden dimension of GCFN is set to $3F$ after GLU to maintain a similar parameter size to the FFN with a hidden size of $4F$. Therefore, while the FFN has parameter size of $8F^2$, GCFN has a slightly larger size of about $9F^2$.

## 3.3 Global and local Transformer for long sequences

Instead of the dual-path model based on chunking, we directly process a long sequence using global and local processing blocks, similar to QDPN [59] or Conformer [27]. In particular, global and local blocks replace inter- and intra-chunk processing, respectively. The design of the blocks follows a Transformer block structure to ensure structural effectiveness [23, 68, 86, 29]. This structure consists of two sub-modules: temporal mixing and frame-wise channel mixing. These modules are stacked together with a pre-norm residual unit [75, 55] and LayerScale [69] to facilitate faster training of deep networks. Also, in all residual units, we apply dropout [64] for regularization.

**Gated convolutional feed-forward network (GCFN)**   Instead of using the conventional feed-forward network (FFN) [23, 72] for channel mixing, we improve it by incorporating temporal Dconv with a small kernel size of 3 and substituting GELU with GLU activation [20] as shown in Figure 4(a). This GCFN can effectively process channel features by considering the adjacent frame context. Several studies also have demonstrated the effectiveness of these enhancements in FFN [63, 87, 77].

**Global Transformer with efficient global attention (EGA)**   In Figure 4(a), the global block consists of an EGA module for temporal mixing and GCFN. The EGA module is based on the MHSA with relative positional encoding [19]. However, to reduce the computation and focus on global information in the attention layer, the downsampled sequence is processed and upsampled back. Sequences $T/2^r$ at all stages $0 \le r \le R - 1$ are downsampled to $T/2^R$, which is equal to the length in the bottleneck. To compensate for downsampling, the upsampled features are multiplied by the gate value obtained from an additional branch with a linear layer and sigmoid function $\sigma$. The simple strategy allows the effective capture of global contexts while maintaining local contexts.

**Local Transformer with convolutional local attention (CLA)**   For the local block, we design a CLA module based on 1D temporal convolution with a large kernel of $K$ in Figure 4(b). Inspired by [27, 84], the CLA module first processes the feature with the Pconv layer and GLU activation to facilitate capturing local contexts attentively. After the temporal Dconv, two Pconv layers are used. They have a hidden dimension of $2F$ and employ BN and GELU activation.

## 3.4 Boosting discriminative learning by multi-loss

The objective function is given as scale-invariant signal-to-noise ratio (SI-SNR) [62, 46] defined as

$$\mathcal{L} = -\sum_{j=1}^{J} \min\left(20\log_{10}\frac{\|\gamma_j \mathbf{s}_j\|_2}{\|\gamma_j \mathbf{s}_j - \hat{\mathbf{s}}_j\|_2}, \tau\right), \tag{1}$$

where $\gamma_j = \hat{\mathbf{s}}_j^T \mathbf{s}_j / \|\mathbf{s}_j\|_2^2$ and $\|\cdot\|_2$ denotes L2-norm. The clipping value of $\tau$ limits the influence of the best training prediction [89, 83]. Notably, the output of the decoder stages can be trained for progressive reconstruction as the feature sequences are already separated in the decoder stages as in the progressive multi-stage strategy [54, 91, 88, 16]. In particular, weight-sharing decoder in each stage

can be trained clearly for discriminative learning with stage-specific separation objective. This multi-loss strategy is also considered to guide intermediate features in audio separation [53, 5, 59, 60, 40].

Therefore, the source signal can be estimated as $\hat{\mathbf{s}}_{j,r} = \mathcal{D}_r(\mathbf{X} \odot \mathbf{M}_{j,r}) \in \mathbb{R}^{1 \times N}$ when $\mathbf{M}_{j,r} \in \mathbb{R}^{F_o \times T}$ is estimated with additional output layers for $\mathbf{L}_{j,r}$ and the nearest upsampling. $\odot$ denotes an element-wise multiplication, and $\mathcal{D}_r(\cdot)$ is an auxiliary audio decoder, which is also additional required with additional output layers. Therefore, we can set the auxiliary objective function as

$$\mathcal{L}_r = -\sum_{j=1}^{J} \min\left(20 \log_{10} \frac{\|\gamma_{j,r}\mathbf{s}_j\|_2}{\|\gamma_{j,r}\mathbf{s}_j - \hat{\mathbf{s}}_{j,r}\|_2}, \tau\right), \tag{2}$$

where $\gamma_{j,r} = \hat{\mathbf{s}}_{j,r}^T \mathbf{s}_j / \|\mathbf{s}_j\|_2^2$. Note that, when calculating the output from intermediate features, we opt for masking instead of direct estimation because the temporal resolutions of the feature sequences are deficient. Then, the multi-loss can be set to $\hat{\mathcal{L}} = (1 - \alpha)\mathcal{L} + \alpha \sum_{r=1}^{R} \mathcal{L}_r / R$. Moreover, we alternatively calculate the intermediate loss $\mathcal{L}_r$ using the magnitude values of $\mathbf{s}_j$ and $\hat{\mathbf{s}}_j$ in the STFT domain as it provided more stable training and no actual separated signals are required from the intermediate outputs.

## 4 Experimental Settings

### 4.1 Dataset

We evaluated our proposed SepReformer on WSJ0-2Mix [30], WHAM! [82], WHAMR! [50], and LibriMix [18], which are popular datasets for monaural speech separation. To ensure generality, the mixtures in the test set were generated by the speakers that were not seen during training. For all the datasets, networks were trained with 4-s-long segments at a 8-kHz sampling rate while the model processes inputs of varying lengths in the evaluation.

**WSJ0-2Mix**   WSJ0-2Mix is the most popular dataset to benchmark the monaural speech separation task. It contains 30, 10, and 5 hours for training, validation, and evaluation sets, respectively. Each mixture was artificially generated by randomly selecting different speakers from the corresponding set and mixing them at a random relative signal-to-noise ratio (SNR) between -5 and 5 dB.

**WHAM!/WHAMR!**   WHAM!/WHAMR! is a noisy/noisy-reverberant version of the WSJ0-2Mix dataset. In the WHAM! dataset, speeches were mixed with noise recorded in scenes such as cafes, restaurants, and bars. The noise was added to get mixtures at SNRs uniformly sampled between -6dB and 3dB, making the mixtures more challenging than those in the WSJ0-2Mix.

**Libri2Mix**   In Libri2Mix dataset, the target speech in each mixture was randomly selected from a subset of LibriSpeech's train-100 [56] for faster training. Each source was mixed with uniformly sampled Loudness Units relative to Full Scale (LUFS) to get a mixture at an SNR between -25 and -33 dB. We used the clean version as in previous studies [9, 42].

### 4.2 Training and model configuration

We trained the proposed SepReformer for a maximum of 200 epochs with an initial learning rate of $1.0e^{-3}$. We used a warm-up training scheduler for the first epoch, and then the learning rate decayed by a factor of 0.8 if the validation loss did not improve in three consecutive epochs. As optimizer, AdamW [43] was used with a weight decay of 0.01, and gradient clipping with a maximum L2-norm of 5 was applied for stable training. All models were trained with Permutation Invariant Training (PIT) [36]. When the multi-loss in Subsection 3.4 was applied, the $\alpha$ was set to 0.4, and after 100 epochs, it decayed by a factor of 0.8 at every five epochs. $\tau$ was set to 30 as in [89]. SI-SNRi and SDRi [73] were used as evaluation metrics. Also, we compared the parameter size and the number of multiply-accumulate operations (MACs) for 16000 samples. The number of heads in MSHA was commonly set to 8, and the kernel size $K$ in the local block was set to 65. Also, we evaluated our model in various model sizes as follows:

- SepReformer-T/B/L: $F = 64/128/256$, $F_o = 256$, $L = 16$, $H = 4$, $R = 4$
- SepReformer-S/M: $F = 64/128$, $F_o = 256$, $L = 8$, $H = 2$, $R = 5$

We used a longer encoder length of $L = 32$ in the Large model when evaluating the WHAMR dataset to account for reverberation. Note that we did not train the models multiple times, as the deviations in the results are negligible below the significant digits. All experiments were conducted on a server with GeForce RTX 3090 × 6.

| Case | MACs (G/s) | Param. (M) | SI-SNRi (dB) |
|---|---|---|---|
| late split+origin dec. | 5.0/18.3 | 2.8/11.6 | 19.0/21.6 |
| late split+large dec. | 9.0/33.7 | 4.9/20.1 | 19.7/22.0 |
| early split+multi dec. | 7.9/29.5 | 4.5/18.4 | 19.8/22.1 |
| early split+shared dec. | 7.9/29.5 | 2.8/11.6 | 21.3/23.1 |
| early split+shared dec.+CS | 10.4/39.8 | 3.5/14.2 | 22.4/23.8 |

(a) Decoder design.

| Case | ML | Param. (M) | SI-SNRi (dB) |
|---|---|---|---|
| late split+origin dec. | | 11.6 | 21.2 |
| late split+origin dec. | ✓ | 13.2* | 21.6 |
| early split+shared dec. | | 11.6 | 22.4 |
| early split+shared dec. | ✓ | 12.2* | 23.1 |
| early split+shared dec.+CS | | 14.2 | 22.6 |
| early split+shared dec.+CS | ✓ | 14.8* | 23.8 |

(b) Effects of multi-loss.

Table 1: **Experimental evaluation of SepRe method** on the WSJ0-2Mix dataset. ML denotes the multi-loss. In (a), all the methods were trained with ML, and the numbers in the left and right of the '/' symbol were obtained for the tiny and base models, respectively. In (b), when ML was used for training, we indicated the numbers of parameters including the additional output layer for an auxiliary output for $\hat{s}_j$, which were denoted with asterisk $^*$. Note that the additional output layers were not required during inference.

| Case | ESSD | CS | ML | Param. (M) | SI-SNRi (dB) |
|---|---|---|---|---|---|
| 1 (origin.) | | | | 5.1 | 15.3 |
| 2 | ✓ | | | 5.4 | 17.5 |
| 3 | ✓ | | ✓ | 5.5 | 17.8 |
| 4 (SepRe) | ✓ | ✓ | | 5.7 | 19.2 |
| 5 (SepRe) | ✓ | ✓ | ✓ | 5.7 | 19.5 |

(a) Conv-TasNet with SepRe method.

| Case | ESSD | CS | ML | Param. (M) | SI-SNRi (dB) |
|---|---|---|---|---|---|
| 1 (origin.) | | | | 26.0 | 20.4 |
| 2 | ✓ | | | 27.1 | 21.3 |
| 3 | ✓ | | ✓ | 27.2 | 22.0 |
| 4 (SepRe) | ✓ | ✓ | | 28.0 | 21.6 |
| 5 (SepRe) | ✓ | ✓ | ✓ | 28.0 | 22.7 |

(b) Sepformer with SepRe method.

Table 2: **Application of SepRe to other networks.** From the original separator of Conv-TasNet and Sepformer, we applied the SepRe method with multi-loss (ML) and evaluated on the WSJ0-2Mix dataset.

## 5 Results

### 5.1 Ablation studies of SepRe method

**Decoder Design** In Table 1(a), we evaluated various decoder structures (See Appendix A for detailed structures) to validate the effectiveness of weight-sharing decoder structure. As shown in Table 1(a), the computations increases about twice by using large decoder in late split or using early split methods. In particular, the model using multiple decoders after an early split yielded a performance comparable to that of using a large decoder after a late split. In contrast, by sharing a decoder after an early split, the separation result increased significantly, suggesting that the ESSD structure effectively discriminates between the separated features. This impact was more noticeable in the tiny models by showing increase of 1.5dB. Applying CS to ESSD improved the performance especially on the tiny model, leading to the SepRe mechanism. Although changing from a late split structure to an ESSD structure can increase computation if the channel size is kept constant, reducing the channel size allows us to still achieve better performance. This adjustment significantly improves computational efficiency in relation to performance. This is particularly evident when comparing the base model with late split to the tiny model with the proposed ESSD + CS (SepRe) structure. The latter model achieves a higher performance, with an SI-SNRi of 22.4 dB compared to 21.6 dB, while using fewer computations and a smaller model size, clearly demonstrating the efficiency of the model architecture.

**Effects of multi-loss** Furthermore, we experimented with the effects of multi-loss on various decoder structures in Table 1(b). Compared to a late split, the case with an early split increased more significantly with multi-loss because an early split structure could be trained with a clearer objective for discriminative learning using an intermediate loss at each stage. In particular, while applying only CS without multi-loss resulted in a marginal improvement, combining CS with multi-loss led to a substantial gain. The results demonstrated that stage-specific objective functions induce each CS-equipped weight-sharing decoder stage to effectively learn simultaneously how to discriminate between and attend to each other. As a result, our proposed SepRe method using ESSD and CS significantly improved separation performance by applying stage-specific objective functions and inducing progressive reconstruction of separated feature sequences.

| $(B_E, B_D)$ | Param. (M) | SI-SNRi (dB) |
|---|---|---|
| (1, 4) | 14.6 | 23.6 |
| (2, 3) | 14.2 | 23.8 |
| (3, 2) | 13.8 | 22.8 |
| (4, 1) | 13.4 | 21.7 |

(a) Depth of encoder-decoder.

| Case | Param. (M) | SI-SNRi (dB) |
|---|---|---|
| MHSA w/ d.s & u.s | 13.9 | 23.3 |
| EGA w/o linear gate | 13.9 | 23.2 |
| EGA | 14.2 | 23.8 |

(b) EGA module design.

| Case | Param. (M) | SI-SNRi (dB) |
|---|---|---|
| FFN [23] | 13.3 | 23.0 |
| FFN w/ Dconv | 13.4 | 23.4 |
| FFN w/ GLU | 14.1 | 23.3 |
| GCFN | 14.2 | 23.8 |

(c) FFN module design.

Table 3: **Ablation studies for unit blocks** on our SepReformer-B on the WSJ0-2Mix dataset. Various configurations of $B_E$ and $B_D$ were evaluated to assess the relative importance of encoder and decoder. Also, we validated the proposed EGA and GCFN modules.

| Separator | Long sequence model | Param. (M) | MACs (G/s) | SI-SNRi (dB) |
|---|---|---|---|---|
| Conv-TasNet [47] | TCN [71] | 5.1 | 10.5 | 15.6 |
| DPRNN [45] | Dual-path + BLSTM | 2.6 | 88.5 | 18.8 |
| SuDoRM-RF [70] | Multi-scale + Convolution | 6.4 | 10.1 | 18.9 |
| Sepformer [66] | Dual-path + Transformer | 26.0 | 86.9 | 20.4 |
| TDANet [42] | Multi-scale + Transformer | 2.3 | 9.1 | 18.5 |
| MossFormer(S) [92] | GAU [33] | 10.8 | 44.0 | 20.9 |
| S4M [7] | Multi-scale + SSM [26] | 3.6 | 38.4 | 20.5 |
| Ours | Global-Local Transformer | 11.9 | 43.1 | 21.3 |
| Ours + U-Net | Multi-scale + Global-Local Transformer | 11.6 | 18.3 | 21.2 |

Table 4: **Comparison with various long sequence models** in speech separation of WSJ0-2Mix. MS denotes multi-scale. For our model, global and local blocks were repeated 22 times with $F = 128$.

## 5.2 Effects of the SepRe method in other networks

To validate the general applicability of the SepRe method, we incorporated the SepRe method with multi-loss into the original separators of Conv-TasNet [47] and Sepformer [66] and conducted experiments on WSJ0-2MIX. The experimental results in Table 2 demonstrated a significant performance improvement when ESSD was applied for both networks. Also, applying CS and multi-loss in addition to the ESSD framework improved the performance further, which confirms the effectiveness of SepRe with multi-loss.

## 5.3 Ablation studies of unit blocks

**Depth of encoder-decoder** In Table 3(a), we experimented the depth of encoder and decoder to determine the optimal configuration in terms of the number of block repetition $B_E$ and $B_D$. Generally, experimental results showed that using more blocks in the reconstruction decoder had a greater impact on performance improvement than in the separation encoder. It demonstrate that it is more important to discriminate the features more elaborately in weight-sharing decoder than to analyze the features in encoder in speech separation. In particular, optimal performance was achieved with $B_E = 2$ and $B_D = 3$, which were used as the common configuration for subsequent experiments.

**EGA module design** Next, we validated our proposed EGA module by ablating its components (see Appendix B for detailed structures) in Table 3(b). First of all, using vanilla MHSA on a long sequence without chunking was infeasible due to the extremely large computational requirements. Therefore, one approach was to perform downsampling before applying MHSA, similar to the method used in QDPN [59]. However, this naive approach had the drawback of losing detailed frame-wise information. Although another consideration was to simply multiply the features to reflect the fine-grained frame-wise information, this method still could not significantly improve performance. In contrast, the optimal performance was achieved by estimating gate values based on a linear layer and a sigmoid function. As a result, it is shown that our proposed global Transformer with EGA module and local Transformer with CLA module have effectively replaced conventional sequence models with smaller computations.

**FFN module design** Also, by improving the design of FFN with Dconv and GLU activation, we could achieve the significant improvement of performance with slight increase of parameters as shown in Table 3(c).

**Comparison with various long sequence models** In Table 4, we evaluated the network by stacking our proposed global-local Transformer blocks to assess the performance of modeling a long sequence.

| System | Params. (M) | MACs (G/s) | WSJ0-2Mix | | WHAM! | | Libri2Mix | |
|---|---|---|---|---|---|---|---|---|
| | | | SI-SNRi (dB) | SDRi (dB) | SI-SNRi (dB) | SDRi (dB) | SI-SNRi (dB) | SDRi (dB) |
| Conv-TasNet [47] | 5.1 | 10.5 | 15.3 | 15.6 | 12.7 | - | 12.2 | 12.7 |
| SuDoRM-RF [70] | 6.4 | 10.1 | 18.9 | - | 13.7 | 14.1 | 14.0 | 14.4 |
| TDANet [42] | 2.3 | 9.1 | 18.5 | 18.7 | 15.2 | 15.4 | 17.4 | 17.9 |
| Sandglasset [38] | 2.3 | 28.8 | 20.8 | 21.0 | - | - | - | - |
| S4M [7] | 3.6 | 38.4 | 20.5 | 20.7 | - | - | 16.9 | 17.4 |
| SepReformer-T | 3.5 | 10.4 | 22.4 | 22.6 | 17.2 | 17.5 | 19.7 | 20.2 |
| SepReformer-S | 4.3 | 21.3 | 23.0 | 23.1 | 17.3 | 17.7 | 20.6 | 21.0 |
| DPRNN [45] | 2.6 | 88.5 | 18.8 | 19.0 | 13.7 | 14.1 | 16.1 | 16.6 |
| DPTNet [9] | 2.7 | 102.5 | 20.2 | 20.3 | 14.9 | 15.3 | 16.7 | 17.1 |
| Sepformer [66] | 26.0 | 86.9 | 20.4 | 20.5 | 14.7 | 16.8 | 16.5 | 17.0 |
| WaveSplit[†] [89] | 29.0 | - | 21.0 | 21.2 | 16.0 | 16.5 | 16.6 | 17.2 |
| A-FRCNN [32] | 6.1 | 125.0 | 18.3 | 18.6 | 14.5 | 14.8 | 16.7 | 17.2 |
| SFSRNet [60] | 59.0 | 124.2 | 22.0 | 22.1 | - | - | - | - |
| ISCIT[†] [51] | 58.4 | 252.2 | 22.4 | 22.5 | 16.4 | 16.8 | - | - |
| QDPN [59] | 200.0 | - | 22.1 | - | - | - | - | - |
| TF-GridNet [79] | 14.5 | 460.8 | 23.5 | 23.6 | - | - | - | - |
| SepReformer-B | 14.2 | 39.8 | 23.8 | 23.9 | 17.6 | 18.0 | 21.6 | 21.9 |
| SepReformer-M | 17.3 | 81.3 | 24.2 | 24.4 | 17.8 | 18.1 | 22.0 | 22.2 |

(a) Comparison of SepReformer to existing models.

| System | Params. (M) | MACs (G/s) | WSJ0-2Mix | | WHAM! | | WHAMR! | |
|---|---|---|---|---|---|---|---|---|
| | | | SI-SNRi (dB) | SDRi (dB) | SI-SNRi (dB) | SDRi (dB) | SI-SNRi (dB) | SDRi (dB) |
| Sepformer [67] | 26.0 | 86.9 | 22.3 | 22.5 | 16.4 | 16.7 | 14.0 | 13.0 |
| WaveSplit[†] [89] | 29.0 | - | 21.0 | 21.2 | - | - | 13.2 | 12.2 |
| SFSRNet [60] | 59.0 | 466.2 | 24.0 | 24.1 | - | - | - | - |
| ISCIT[†] [51] | 58.4 | 252.2 | 24.3 | 24.4 | 16.9 | 17.2 | - | - |
| QDPN [59] | 200.0 | - | 23.6 | - | - | - | 14.4 | - |
| Mossformer(L) [92] | 42.1 | 86.1 | 22.8 | - | 17.3 | - | 16.3 | - |
| Mossformer2(L) [93] | 55.7 | - | 24.1 | - | 18.1 | - | 17.0 | - |
| Separate And Diffuse [48] | - | - | 23.9 | - | - | - | - | - |
| zSepReformer-L | 59.4 | 155.5 | 25.4 | 25.6 | 18.4 | 18.7 | 17.2 | 16.0 |

(b) Comparison of SepReformer-L to existing large models with DM.

Table 5: **Evaluation on various benchmark dataset of WSJ0-2MIX, WHAM!, WHAMR!, and Libri2Mix**. "†" denotes that the networks use additional speaker information.

Note that we did not apply the ESSD structure and multi-loss to our separator in this experiment. We could observe that the network based on dual-path sequence models requires high computation resources in terms of MACs while multi-scale sequence models are more efficient. The recently proposed Mossformer based on efficient gate attention unit (GAU) mechanism [33] showed improved performance with relatively smaller computations compared to the networks with dual-path model. In particular, the proposed model showed improved separation performance with similar MACs, which demonstrated the capacity as a model for a long sequence. It also suggested that the proposed block can sufficiently replace the dual-path models with fewer computations. Furthermore, by combining the U-Net structure into global-local Transformer blocks, the network became more efficient with the similar separation performance.

## 5.4 Comparison with existing models

Finally, we compared our SepReformer models with existing separation models on various benchmark datasets in Table 5. Although we evaluated SepReformer trained with standard pairs from the training set in Table 5(a), SepReformer-L was trained with dynamic mixing (DM) [89, 66] for data augmentation and compared to other existing large models with DM in Table 5(b). When traind with DM, we set an initial learning rate of $2.0e^{-4}$ and fixed during first 50 epoch. In Table 5(a), with almost the smallest computational loads in terms of MACs, our tiny model showed the best performance except for TF-GridNet in the WSJ0-2Mix dataset which was a powerful model recently proposed. It demonstrated the efficiency of the SepRe method in speech separation. Also, SepReformer-M

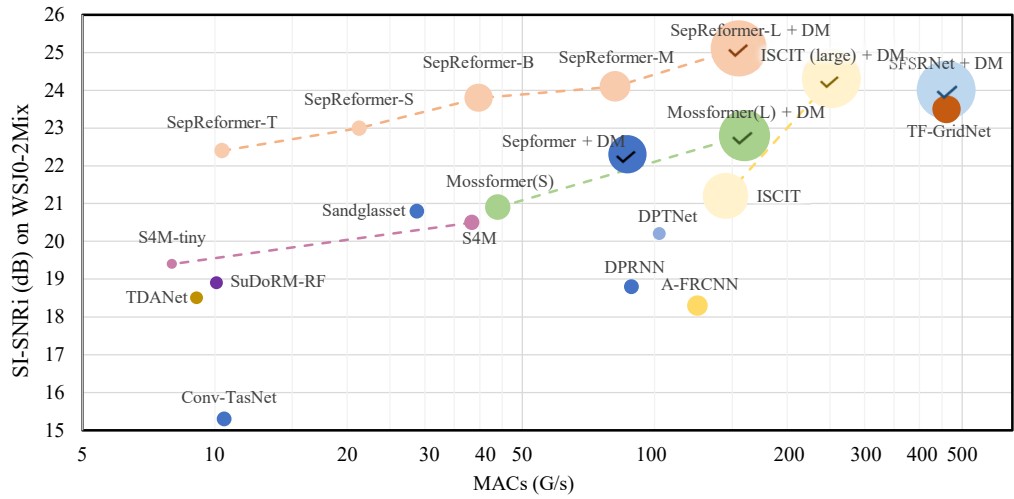

Figure 5: **Si-SNRi results on WSJ0-2Mix versus MACs (G/s)** for the conventional methods and the proposed SepReformer. The check mark in the circle indicates the use of DM method for training. The radius of circle is proportional to the parameter size of the networks.

without DM in Table 5(a) showed competitive separation performance on WSJ0-2Mix compared to the large models with data augmentation in Table 5(b). In particular, SepReformer-L with DM achieved a state-of-the-art performance of 25.4 dB SI-SNRi on the WSJ0-2Mix dataset, demonstrating a significant improvement over other conventional methods.

In Table 5(a), the smallest SepReformer-T among the proposed models even showed significant improvements on WHAM! and Libri2Mix datasets compared to the conventional methods. It suggested that the proposed SepRe method can be efficiently applied to a speech separation task in general. Also, SepReformer-L with DM showed the SOTA performance on WHAM! and WHAMR! datasets, as well as WSJ0-2Mix, which demonstrated that the proposed method can be trained effectively in a large model. Figure 5 compares the separation performance of various existing methods in terms of SI-SNRi versus MACs on WSJ0-2MIX. From the figure, we can observe the significant effectiveness of the proposed SepReformer in the speech separation task with high computational efficiency. Especially, it is noteworthy that the SepReformer-T models outperformed the conventional Sepformer trained with DM with 10 times smaller computations.

## 6 Conclusion

In this work, we introduced the SepRe method, in which the asymmetric encoder and decoder perform separation and reconstruction, respectively. The encoder analyzes and separates a feature sequence, and the separated sequences are reconstructed by a weight-sharing network and a cross-speaker network. We demonstrated that the SepRe method can be applied to conventional separators in general and utilizing multi-loss significantly improves the performance. Moreover, we replaced the dual-path model with presented global and local Transformer blocks to address a long sequence. The separator using the presented unit blocks has shown enhanced separated results efficiently, and combining a U-Net structure to exploit the multi-scale sequence model has further increased the efficiency. Finally, not only did our presented SepReformer outperform the most conventional methods in speech separation even with almost the smallest computational resources, but our large models achieved SOTA performance with large margins compared to the conventional models on various speech separation datasets.

**Limitations and future work.** Our study focuses on 2-speaker mixture situation to assess our models in various model sizes and in the extensive datasets including noise and reverberation. Consequently, we believe that further investigation is needed to validate for more than 2-speaker mixture scenarios. Also, an important future direction is to separate mixtures for an unknown number of speakers as it is impractical to assume that the number of speakers to be separated is known in advance. Finally, although we experimentally validated our SepRe method, we believe that further investigation is necessary to figure out its underlying mechanism.

## Acknowledgments and Disclosure of Funding

This work was supported in part by Institute of Information & communications Technology Planning & Evaluation (IITP) grant funded by the Korea government (MSIT) (RS-2022-II220989(2022-0-00989), Development of Artificial Intelligence Technology for Multi-speaker Dialog Modeling), and in part by the National Research Foundation of Korea (NRF) and the Commercialization Promotion Agency for R&D Outcomes (COMPA) grant funded by the Korea government (MSIT) (RS-2023-00237117).

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

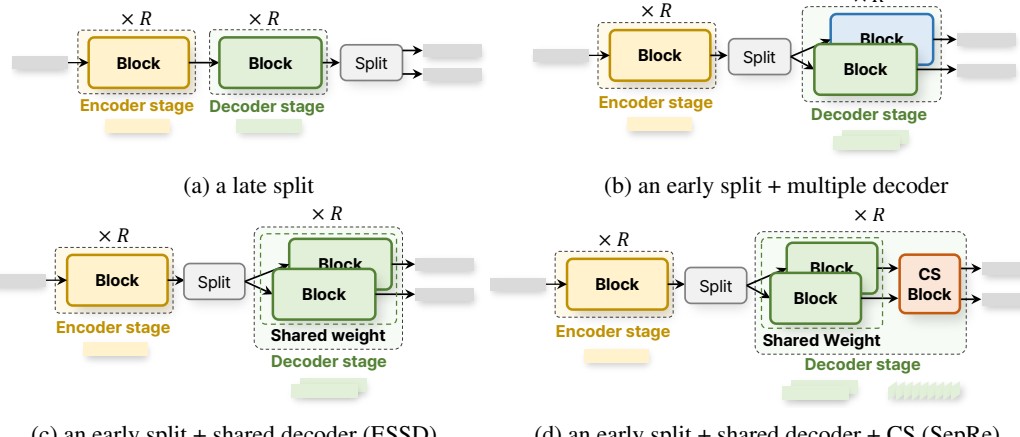

(a) a late split

(b) an early split + multiple decoder

(c) an early split + shared decoder (ESSD)

(d) an early split + shared decoder + CS (SepRe)

Figure 6: **Block diagrams of various decoder designs experimented in Table 1 of subsection 5.1.** In all cases, the encoder and decoder consists of $R$ stages and the blocks were stacks of global and local Transformer block in our cases.

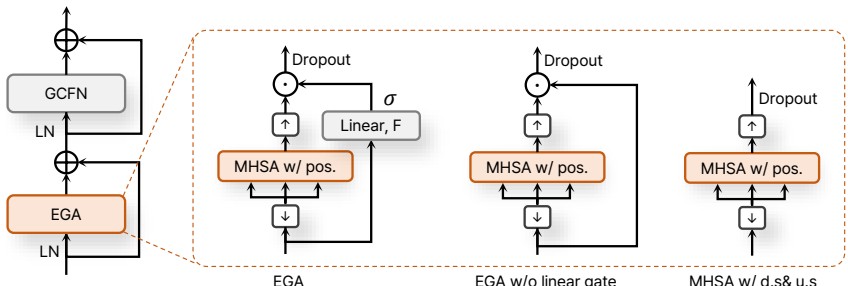

Figure 7: The block diagram of ablation studies for EGA in Table 3(b).

## Appendix / supplemental material

This appendix is organized as follows:

- Appendix A describes the various decoder designs in Table 1(a).
- Appendix B illustrates the cases of ablating EGA module in Table 3(b)
- Appendix C experiments by comparing two speaker split layer designs.
- Appendix D interprets the discriminative learning mechanism in reconstruction decoder.
- Appendix E evaluates perceptual quality on WHAMR dataset.
- Appendix F experiments on real-recorded reverberant mixture for overlapped speech recognition.

## A    Architecture of various decoder design

In Figure 6, the block diagrams of various decoder designs are illustrated, which were experimented on in Table 1. In Table 1(a), the case of a late split, corresponding to the first and second rows, processes a single feature sequence in both the encoder and decoder, forming a symmetric encoder-decoder structure. In contrast, the case of an early split with multiple decoders, shown in Figure 6(b), has each decoder block processing the separated sequences from the encoder. Thus, Figure 6(b) can be seen as a special case of a late split with a large decoder of $2F$, where the large decoder performs group operation with the number of groups equal to the number of speakers $J$. On the other hand, in the ESSD method illustrated in Figure 6(c), the decoder shares weights to process the early split feature sequences. Therefore, even without interaction between the separated feature sequences, the decoder can learn discriminative characteristics by sharing the weights. Furthermore, in the proposed SepRe method shown in Figure 6(d), the decoder learns to attend to each other using the CS block to additionally recover the deviated elements.

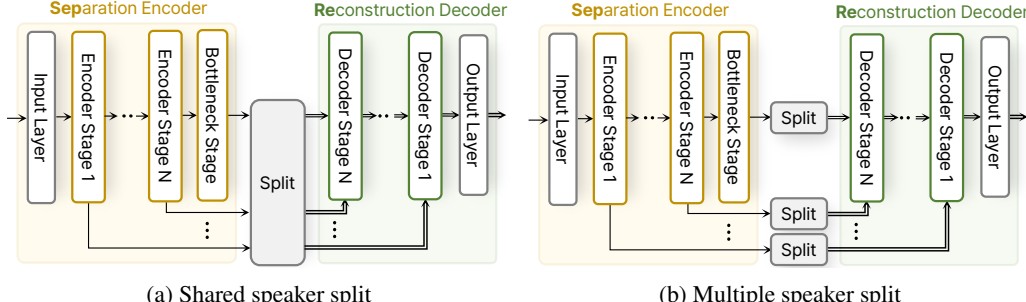

(a) Shared speaker split        (b) Multiple speaker split

Figure 8: The block diagram of shared and multiple speaker split layer in SepReformer architecture.

| Speaker Split | Params. (M) | SI-SNRi (dB) | SDRi (dB) |
|---|---|---|---|
| Weight-shared layer | 55.3 | 25.2 | 25.3 |
| Multiple layer | 59.4 | 25.1 | 25.2 |

Table 6: Comparison of shared and multiple speaker split layer based on SepReformer-L with DM on the WSJ0-2Mix, WHAM!, and WHAMR! dataset.

## B    Illustration of ablation for EGA module

In Figure 7, we drew a block diagram of ablation studies for EGA in Table 3(b). In the fisrt row of Table 3(b), MHSA is simply performed with downsampling and upsampling to reduce the sequence length. In the second row, the upsampled output sequences of MHSA with downsampling is multiplied to the input features before downsampling in order to reflect detailed temporal information. However, simply multiplying the input features does not improved the performances. To reflect the frame-level details to the upsampled feature, we added the linear layer and sigmoid function to make gate values, leading to our proposed EGA module.

## C    Comparison of shared and multiple split layers

In designing the SepReformer architecture, a key consideration is whether to use a shared speaker split layer across all feature stages as illustrated in Figure 8(a), or to implement distinct speaker split layers for each stage as in Figure 8(b). Our hypothesis is that a shared speaker split layer would enable the reconstruction decoder to process consistently separated feature sequences. Conversely, assigning a unique split layer to each stage in the separation encoder could account for stage-specific feature variations, as shown in Figure 8(b). Therefore, we conducted comparative experiments on these two configurations.

Upon comparing the two approaches of the multiple and the shared split layer, interestingly, different tendencies are observed depending on the type of mixture, whether it was an anechoic clean mixture or a noisy or, furthermore, noisy-reverberant mixture, in terms of SI-SNRi and SDRi performance (as shown in Table 6). For the WSJ0-2Mix dataset, which comprises simple anechoic mixtures, the shared split layer method produced results that were comparable to, or slightly better than, those of the multiple split layer approach, in line with our hypothesis. However, for the WHAM dataset, containing noisy mixtures, the multiple split layer approach demonstrated improved results, with the performance gap still sustained on the WHAMR dataset, which includes noisy and reverberant mixtures. Based on these findings, we opted for the shared split layer approach in most cases, as it provided competitive results while reducing the model's parameter count. On the other hand, for the SepReformer-L with DM in Table 5(b), we employed multiple split layers to enhance performance, particularly on the WHAM and WHAMR datasets.

## D    Visualization of discriminative learning

To analyze the roles of each transformer block within the shared decoder of the ESSD framework with multi-loss application, we calculate the cosine similarities between the separated features from $Z1$ to $Z4$ as shown in Figure 9. First, we observe that $Z1$ has higher cosine similarity compared to the others, indicating that the separated features initially share similar characteristics before being processed by the weight-sharing blocks. However, after the weight-sharing global block, the similarities in $Z2$ are generally lower compared to $Z1$. This result suggests that the weight-sharing global blocks capture discriminative characteristics, making the features more dissimilar in terms of cosine similarities. The further decrease in similarity in $Z3$ compared to

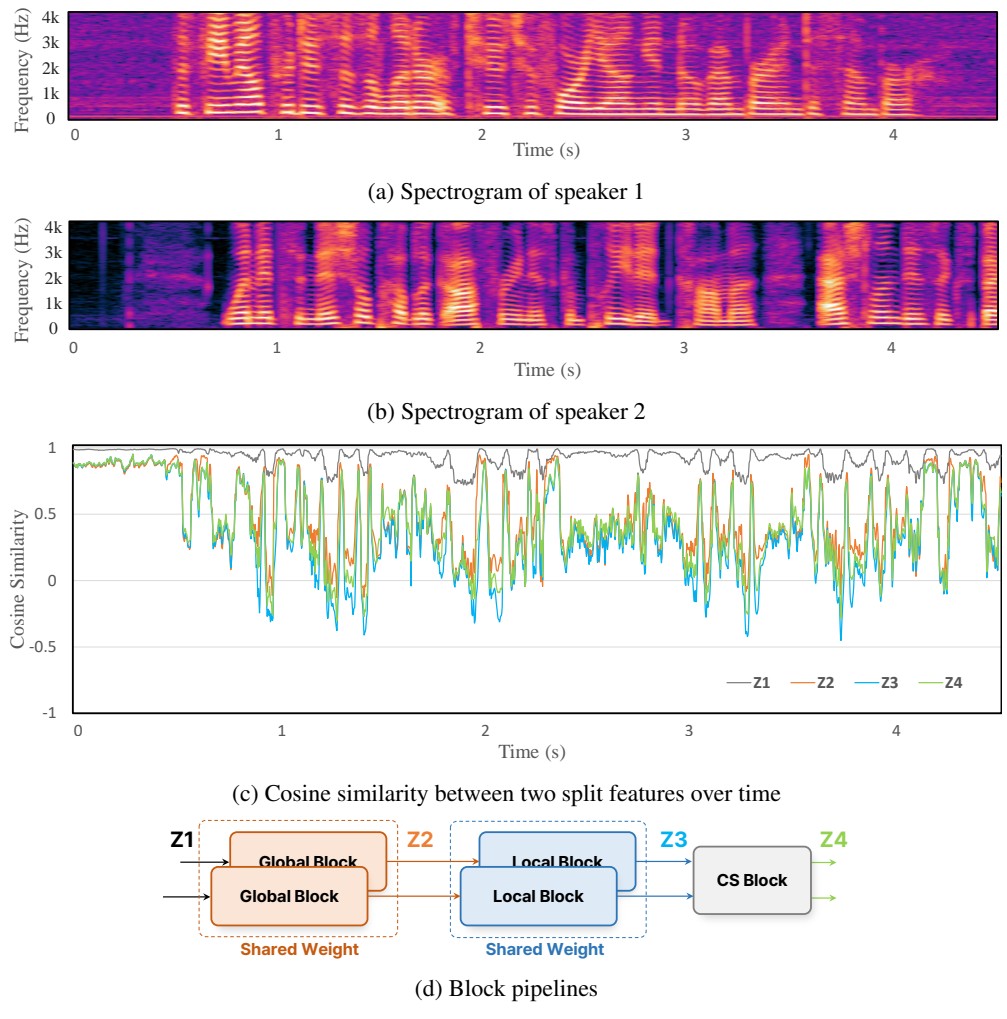

(a) Spectrogram of speaker 1

(b) Spectrogram of speaker 2

(c) Cosine similarity between two split features over time

(d) Block pipelines

Figure 9: **Plot of cosine similarities for the two separated features in the first decoder stage** using a sample mixture in WSJ0-2Mix dataset.

| Separator | SI-SNRi (dB) | SDRi (dB) | PESQ | eSTOI |
|---|---|---|---|---|
| No Processing | -6.1 | -3.5 | 1.41 | 0.317 |
| TF-GridNet [78] | 10.6 | 11.7 | 2.75 | 0.793 |
| SepReformer-L | 11.0 | 12.85 | 2.78 | 0.798 |

Table 7: Perceptual evaluation by PESQ and eSTOI on WHAMR! dataset.

**Z2** and **Z1** demonstrates the effect of the weight-sharing structure of the local block. As features pass through subsequent local blocks, more region-specific traits are refined. The local block, which handles these localized characteristics, further enhances the distinctiveness of the local features, resulting in decreased similarity.

In contrast, features processed through the CS block exhibit increased similarity, unlike the weight-sharing structure. This increase in similarity can be understood as the separated features attending to and becoming more similar to each other, as the CS structure is designed to cross-reference information between the speech features. During this process, the CS block preserves distinct features and restores degraded information by leveraging mutual information. As the weight-sharing structure emphasizes unique characteristics, the split features can deviate from the original characteristics of the speech within the same frames, where the influence of each speaker's information is similar. This deviation occurs because emphasizing features in such frames can increase interference from other speakers' speech components, potentially distorting and degrading these features. Therefore, the CS block after the weight-sharing block is expected to recover the deviated features by attending to each other within the frames.

| Separator | Overlap Ratio (%) | | | | | |
|---|---|---|---|---|---|---|
| | 0S | 0L | 10 | 20 | 30 | 40 |
| No Processing | 11.8 | 11.7 | 18.8 | 27.2 | 35.6 | 43.3 |
| BLSTM [13] | 15.8 | 14.2 | 18.9 | 25.4 | 31.6 | 35.5 |
| Conformer [10] | 12.9 | 12.2 | 15.1 | 20.1 | 24.3 | 27.6 |
| DPRNN [47] | 10.6 | 10.4 | 12.7 | 16.6 | 20.8 | 23.5 |
| SepReformer-B | 9.6 | 10.2 | 10.6 | 12.5 | 13.8 | 16.1 |

Table 8: WERs (%) of utterance-wise evaluation on the LibriCSS dataset for the baseline without any processing for input data acquired at the center microphone and separation by LSTM, Conformer, DPRNN, and the proposed SepReformer.

## E    Evaluation of speech perception measures

While SI-SNRi is often the primary metric for speech separation tasks, it's important to also report perceptual metrics like PESQ and eSTOI, especially for datasets like WHAMR! that include noise and reverberation. Table 7 shows that SepReformer-L demonstrates effective performance compared to TF-GridNet [78], with a PESQ of 2.79 and an eSTOI of 0.799, slightly improving upon TF-GridNet's scores of 2.75 and 0.793. No Processing baseline performs poorly, as expected, with a PESQ of 1.41 and eSTOI of 0.317, reflecting low perceptual quality and intelligibility without any enhancement. TF-GridNet shows notable results with SI-SNRi of 10.6 dB, while SepReformer-L shows slightly better performance across both perceptual and signal-level metrics. This suggests that SepReformer-L is also effective in maintaining perceptual quality and intelligibility in challenging environments, making it a relevant model for real-world applications.

## F    Evaluation on real-recorded mixture

To further validate the applicability of the proposed model in real-world environments, we evaluated word error rate (WER) on the LibriCSS [13] dataset using baseline speech recognition model. The dataset consists of recordings that simulate real meeting scenarios, allowing us to assess the separation performance of model as a pre-processing step for overlapped speech recognition. The LibriCSS data set enables the evaluation of varying levels of overlap, ranging from 0% to 40%, which helps not only in measuring separation performance but also in verifying the model's ability to preserve speech quality when overlap is minimal. Accordingly, the proposed model was trained using source-aggregated SDR (SA-SDR) [74] instead of conventional averaged SDR, with varying overlap ratios to account for different levels of overlap during training. For comparison, we additionally trained and evaluated the DPRNN model as a competitive baseline. Since LibriCSS dataset does not provide a training set, we trained the model using LibriSpeech [56] for speech source with noise simulated using colored noise and reverberation based on room impulse response (RIR) simulations using gpuRIR [22]. Although the evaluation dataset includes both reverberation and moderate background noise, we excluded dereverberation during training to ensure stability. Thus, the model was trained to focus solely on speech separation and noise reduction.

Referring to Table 8, we can observe that the word error rate (WER) increases as the overlap ratio (OV) rises for the unprocessed input. Even in cases with no overlap, the baseline results show around 10% WER due to inherent distortions from reverberations and background noises. The BLSTM [13] and Conformer [10] models, which are based on STFT and real-valued masking, show some improvement at higher OV ratios, but their performance tends to degrade compared to the input when the OV is low, indicating unstable results. This instability could be attributed to the negative effects of attempting to remove reverberation as well. In contrast, the results of the DPRNN model demonstrate consistent improvements over the unprocessed input across all overlap ratios. The proposed SepReformer model further improves performance. This suggests that the speech separation was performed effectively. However, since reverberation was not removed, the results show less improvement in the OV0 condition, which can be interpreted as the model successfully preserving the original speech without unnecessary distortion.

