# OpenReview forum: "Separate and Reconstruct: Asymmetric Encoder-Decoder for Speech Separation"
_NeurIPS.cc/2024/Conference — NeurIPS 2024 poster_

### Official Review · Reviewer_y9eH · 2024-06-24

**Soundness:** 3
**Presentation:** 1
**Contribution:** 3
**Rating:** 5
**Confidence:** 3

**Summary:**

This paper proposes a novel transformer based architecture, sepreformer, for speech separation based on a siamese decoder network that operates on separated speech signals in the encoded space.

**Strengths:**

A major strength of the paper is it's state of the art performance on a variety of datasets, showing as high as 25dB on the WSJ 2 mix. In addition, sepreformer is quite computationally efficient when compared with other baseline methods. The supplementary materials include wav files which show the fidelity of the reconstructed examples. The ablation studies provide good insights into the contributions of different component, such as the depth of the encoder decoder. Details of the method are provided with high degree of reproducibility and description.

**Weaknesses:**

The first major weakness is the presentation quality of the paper. Starting with the abstract, which goes right into a discussion of feature length and computation without setting up the problem and overview. I think a lot of the high level picture is missing, especially related to the particular insights that help this method work better.

Beyond the presentation, a main concern I have of this method is extending to beyond 2 sources. The experiments focus on two-speaker separation. It's unclear how well the method would generalize to scenarios with more than two speakers, which is an important real-world use case. It's also particularly important here because the authors propose separating the sources first and then using a siamese network for decoding the features back to speech. With more than 2 speakers, it's quite likely that the method would have a harder time as the features for each source get separated out earlier in the process.

The experiments and comparisons primarily focuses  on SI-SNRi. I'd like to see other metrics like PESQ or STOI considered as SNR does not always represent perceptual quality.

**Questions:**

Are there any experiments conducted on more than 2 speakers? Are there any metrics considered beyond SNR?

---

> ### Author Rebuttal · Authors · 2024-08-05
>
> **Q1: The first major weakness is the presentation quality of the paper. Starting with the abstract, which goes right into a discussion of feature length and computation without setting up the problem and overview. I think a lot of the high level picture is missing, especially related to the particular insights that help this method work better.**
>
> A1: Thank you for your feedback on the presentation quality of the paper. We recognize that the abstract lacks a detailed problem formulation and overview, which can make it challenging for readers to grasp the context and significance of our work.
>
> In the revised paper, we will include an earlier explanation of the TasNet concept in the abstract to enhance clarity. Also, we agree that TasNet overview should be more detailed. Therefore, we will expand on the concept of TasNet and the task of speech separation in the introduction.
>
> Moreover, we agree that we did not provide sufficient explanation with high-level picture why proactively expanding the speaker dimension with early split (ESSD method) is more advantageous than the late split. To show effectiveness of the proposed method, we will clarify the task and challenge of speech separation, emphasizing that it requires generating more information than what is provided in the original input data. Also, we will clarify that
>
> 1. generating the separated feature at once just before the audio decoder (late split) can be overwhelming for the separator.
> 2. and, Instead, splitting the feature early and using a reconstructing decoder can ease the task for the encoder of separator.
>
> Please refer to Figure 6 in the appendix to demonstrate these points.
>
> By clarifying these points, we aim to improve the presentation quality clarity and make the high-level picture and insights of our method more accessible to the readers.
>
> **Q2: Beyond the presentation, a main concern I have of this method is extending to beyond 2 sources. The experiments focus on two-speaker separation. It's unclear how well the method would generalize to scenarios with more than two speakers, which is an important real-world use case. Are there any experiments conducted on more than 2 speakers?**
>
> A2: We appreciate your concern about extending our method to more than two speakers. We agree that demonstrating effectiveness in multi-source scenarios is crucial for real-world applicability.
>
> While our current paper focuses on two-speaker separation, we recognize the importance of exploring multi-speaker scenarios. We plan to include detailed experimental results and discussions on this topic in our future research, as noted in the conclusion of Section 6. We are confident in the scalability of our method.
>
> Our ESSD mechanism is designed to handle multiple speakers by naturally extending the feature dimension and increasing computation in proportion to the number of sources. This contrasts with conventional methods, which do not adjust the feature dimension based on the number of speakers, often limiting their effectiveness as the number of sources increases.
>
> We are currently working on experiments using the WSJ-{3,4,5}MIX datasets, which include scenarios with more than two sources. Preliminary results are promising; for example, our SepReformer-B achieved an SI-SNRi of 23.5 dB on WSJ-3MIX, nearing state-of-the-art performance. We anticipate further improvements with larger models and additional research.
>
> **Q3: The experiments and comparisons primarily focuses on SI-SNRi. I'd like to see other metrics like PESQ or STOI considered as SNR does not always represent perceptual quality. Are there any metrics considered beyond SNR?**
>
> A3: We primarily focused on SI-SNRi because most studies on speech separation (unlike speech enhancement) evaluated separation performance using only signal-level metrics of SNR, particularly for simple instantaneous mixtures without background noise and reverberation. However, we agree that reporting additional metrics such as PESQ and STOI for datasets like WHAMR!, which include noise and reverberation, would be valuable. The table below shows PESQ and eSTOI results, demonstrating competitive performance compared to the recent powerful TF-GridNet model [1].
>
> |  | SI-SDR (dB) | SDR (dB) | PESQ | eSTOI |
> | --- | --- | --- | --- | --- |
> | Unprocessed | -6.1 | -3.5 | 1.41 | 0.317 |
> | TF-GridNet[1] | 10.6 | 11.7 | 2.75 | 0.793 |
> | SepReformer-L | 11.0 | 12.5 | 2.77 | 0.796 |
>
> Therefore, we will include the PESQ and STOI for the WHAMR! dataset in the appendix.
>
> Additionally, evaluating speech recognition accuracy for separated speech using real data like LibriCSS [2] would be meaningful for real-world applications. We will include WER evaluation results for the LibriCSS dataset in the appendix. Although we cannot provide WER results now due to the need for additional training with background noise and reverberation, we include perviously evaluated comparison results of an early version of SepReformer with DPRNN for reference:
>
> | WER on LibriCSS |  |  |  |  |  |  |
> | --- | --- | --- | --- | --- | --- | --- |
> | Condition | Overlap Ratio in % |  |  |  |  |  |
> |  | 0S | 0L | 10 | 20 | 30 | 40 |
> | Oracle | 4.9 | 5.1 | - | - | - | - |
> | Input | 11.8 | 11.7 | 18.8 | 27.2 | 35.6 | 43.3 |
> | DPRNN | 10.6 | 10.4 | 12.7 | 16.6 | 20.8 | 23.5 |
> | Early version of SepReformer | 9.8 | 10.1 | 10.9 | 12.5 | 14.4 | 17.5 |
>
> Note that the performance of SepReformer is expected to be better than shown.
>
> ---
>
> [1] Z. -Q. Wang, S. Cornell, S. Choi, Y. Lee, B. -Y. Kim and S. Watanabe, "TF-GridNet: Integrating Full- and Sub-Band Modeling for Speech Separation," in IEEE/ACM Transactions on Audio, Speech, and Language Processing, vol. 31, pp. 3221-3236, 2023
>
> [2] Z. Chen et al., "Continuous Speech Separation: Dataset and Analysis," ICASSP 2020 - 2020 IEEE International Conference on Acoustics, Speech and Signal Processing (ICASSP), Barcelona, Spain, 2020, pp. 7284-7288

---

### Official Review · Reviewer_UNNK · 2024-07-03

**Soundness:** 3
**Presentation:** 3
**Contribution:** 2
**Rating:** 6
**Confidence:** 5

**Summary:**

This paper presents a novel approach to time-domain speech separation, departing from the conventional chunk-based dual-path processing. The authors introduce an asymmetric encoder-decoder architecture, where the encoder analyzes features and splits them based on the number of speakers. A Siamese decoder reconstructs the separated sequences, learning to discriminate features without explicit speaker information. The use of global and local Transformer blocks for long-sequence processing eliminates the need for chunking, contributing to a more efficient and effective model.

**Strengths:**

1. The paper introduces an innovative asymmetric encoder-decoder framework for time-domain speech separation, deviating from the standard chunk-based dual-path models.
2. Efficient Feature Discrimination: The Siamese decoder enables the model to learn to discriminate features directly without relying on explicit speaker information, leading to a more streamlined and potentially more robust separation process.
3. The proposed model achieves good performance on benchmark datasets while requiring significantly less computation than previous approaches. This demonstrates the potential for this method to be applied in real-world applications where computational resources may be limited.

**Weaknesses:**

The paper presents some interesting ideas, but their novelty and significance are questionable:

1. Transformer Usage: While the use of Transformer blocks is highlighted, similar architectures have been successfully employed in previous works like Sepformer, raising questions about the uniqueness of this contribution.
2. Limited Evaluation: The experimental results primarily focus on two-speaker separation, which is considered a relatively solved problem in the current state-of-the-art. The absence of evaluations on more challenging scenarios with three or more speakers limits the generalizability and impact of the findings.
3. Incomplete Comparison: The paper's claims of achieving state-of-the-art results are undermined by the lack of comparison with other important papers in the field. Notably, models like the "DIFFUSION-BASED SIGNAL REFINER FOR SPEECH SEPARATION" have reported superior performance (SI-SDR of 23.1dB), raising concerns about the validity of the SOTA claim.

**Questions:**

--

**Limitations:**

--

---

> ### Author Rebuttal · Authors · 2024-08-05
>
> **Q1: Transformer Usage: While the use of Transformer blocks is highlighted, similar architectures have been successfully employed in previous works like Sepformer, raising questions about the uniqueness of this contribution.**
>
> A1: Thank you for your comment. We agree that the usage of Transformer blocks is now very common in the speech separation field. Sepformer model primarily replaces the LSTM in a dual-path structure, similar to DPRNN, with classic Transformer blocks. However, our proposed method does not use a dual-path approach. Instead, we designed Global and Local Transformers with EGA and CLA, replacing the simple multi-head self-attention (MHSA) to handle global and local modeling more effectively. Additionally, we have enhanced the vanilla FFN by integrating it with a gating mechanism to create the GCFN, which efficiently models adjacent frames. Therefore, our approach is substantially different from the straightforward application of Transformer, and we believe these enhancements represent significant contributions. Experimental results of Table3(b), Table3(c), and Table 5 show the effectiveness of proposed Local and Global Transformer alone, without SepRe method.
>
> **Q2: Limited Evaluation: The experimental results primarily focus on two-speaker separation, which is considered a relatively solved problem in the current state-of-the-art. The absence of evaluations on more challenging scenarios with three or more speakers limits the generalizability and impact of the findings.**
>
> A2: We appreciate your concern regarding the extension of our method to scenarios with more than two speakers. Although we focused on experiments using datasets including noisy and noisy-reverberant mixture to show our model’s generalizability in this paper, we are also confident that our ESSD mechanism will perform effectively with even more than two speakers compared to the conventional methods.
>
> Our ESSD method naturally extends the feature dimension and requires increased computation proportional to the number of sources to be separated. Meanwhile, conventional methods do not adjust the feature dimension based on the number of speakers. This limitation in the conventional methods is significant because the separation difficulty and network requirements vary with the number of sources.
>
> To validate this, we are conducting on experiments on the WSJ-{3,4,5}MIX datasets, which include scenarios with more than two sources. We expect that our model will still show SOTA performance in these multi-source scenarios. Specifically, in the WSJ-3MIX, our SepReformer-B showed an SI-SNRi of 23.5 dB, which is comparable to the SOTA performance, and we expect that the score will increase further for a larger model. Although these results were not included in the paper, the experimental results and their discussion will be included in our future work, as mentioned in the conclusion of Section 6.
>
> **Q3: Incomplete Comparison: The paper's claims of achieving state-of-the-art results are undermined by the lack of comparison with other important papers in the field. Notably, models like the "DIFFUSION-BASED SIGNAL REFINER FOR SPEECH SEPARATION" have reported superior performance (SI-SDR of 23.1dB), raising concerns about the validity of the SOTA claim.**
>
> A3: Thank you for your comment. We acknowledge the importance of comprehensive comparisons in validating state-of-the-art (SOTA) claims.
>
> The paper you mentioned [1] evaluates the separation performance based on the WSJ0-2MIX dataset, reporting an SI-SDRi of 23.1 dB. To address your concern, we have indeed compared our results with competitive models in Table 4 of our paper.
>
> - Our SepReformer-S model, which has a smaller model size and lower computational requirements than many recent powerful networks, shows an SI-SDRi of 23.0 dB, which is comparable to the performance of the diffusion-based model you mentioned.
> - Furthermore, our SepReformer-L model demonstrates an SI-SDRi of 25.1 dB, which to the best of our knowledge, represents state-of-the-art performance with a significant margin over existing models.
>
> We recognize the diffusion-based model’s performance as reported in [1] and [2], with results of 23.1 dB and 23.9 dB, respectively. However, it is important to note that these models did not include evaluations on noisy and noisy-reverberant datasets, and they did not provide details on model size and computational resources. Therefore, we chose not to include them in our comparison table. Nevertheless, we agree that it is still valuable to provide the results of the diffusion-based model in Table 4 according to your comment. Therefore, we will include the result of [2] which shows more competitive result than [1].
>
> Thank you again for your valuable feedback.
>
> [1] Hirano, Masato, et al. "Diffusion-based Signal Refiner for Speech Separation." *arXiv preprint arXiv:2305.05857,* 2023.
>
> [2] Lutati, Shahar, Eliya Nachmani, and Lior Wolf. "Separate and diffuse: Using a pretrained diffusion model for better source separation." The Twelfth International Conference on Learning Representations. 2024.

---

### Official Review · Reviewer_aFMn · 2024-07-12

**Soundness:** 3
**Presentation:** 3
**Contribution:** 3
**Rating:** 6
**Confidence:** 4

**Summary:**

The paper proposes SepReformer, an efficient time-domain separation network. The model is an encoder-decoder architecture that splits the output features of the encoder based on the number of speakers before feeding them to the decoder. Both encoder and decoder networks are comprised of transformer blocks that capture global and local characteristics of the signal in different time-scales. The proposed approach achieves state of the art results in 3 well-established datasets.

**Strengths:**

1. Paper well-written and easy to follow.
2. Efficient architecture that produces state of the art results.
3. Thorough experimentation and ablation analysis of their approach.

**Weaknesses:**

1. Testing on clean data. It would be interesting to see how the proposed model performs on noisy datasets.
2. Evaluation of the approach to the two-speaker separation problem.

**Questions:**

1. How does the early split strategy affect the overall computational efficiency compared to late split and other conventional methods?

2. Why was the alpha value set to 0.4, and have you experimented with other values to determine the optimal setting?

3. How does the model handle varying lengths of input sequences, and what is the maximum sequence length it can effectively process?

4. Are there any limitations or failure modes of the proposed method that have been identified during experimentation?

**Limitations:**

1. The paper does not discuss potential failure modes or limitations observed during experimentation,

---

> ### Author Rebuttal · Authors · 2024-08-05
>
> **Q1: Testing on clean data. It would be interesting to see how the proposed model performs on noisy datasets.**
>
> A1: Thank you for your comment.  We performed experiments on noisy and noisy-reverberant datasets in Table 4 to show our models generalizability power. Please refer to it.
>
> **Q2: Evaluation of the approach to the two-speaker separation problem.**
>
> A2: We appreciate your concern regarding the extension of our method to scenarios with more than two speakers. However, we are confident that our ESSD mechanism will perform effectively even with more speakers compared to conventional methods.
>
> Our ESSD method naturally extends the feature dimension and requires increased computation proportional to the number of sources to be separated. This approach is more reasonable compared to conventional methods, which do not adjust the feature dimension based on the number of speakers. This limitation in conventional methods is significant because the separation difficulty and network requirements vary with the number of sources.
>
> To validate this, we are conducting experiments on the WSJ-{3,4,5}MIX datasets, which include scenarios with more than two sources. We expect our model to show SOTA performance in these multi-source scenarios. Specifically, in WSJ-3MIX, our SepReformer-B showed  an SI-SNRi of 23.5 dB, which is near SOTA performance, and we believe the performance will increase further for larger models. Although these results were not included in the paper, the experimental results and their discussion will be included in our future work, as mentioned in the conclusion of Section 6.
>
> **Q3: How does the early split strategy affect the overall computational efficiency compared to late split and other conventional methods?**
>
> A3: Thank you for your valuable comment. First, even with the same number of parameters, using the ESSD structure will approximately increase the computation by factor of the number of speakers in the decoder. Therefore, simply changing a late split structure to an ESSD structure can potentially increase the computation significantly if the channel size is kept constant. Adding a Cross-Speaker module would further increase the computation slightly.
>
> However, with the ESSD structure, we can reduce the channel size while still achieving higher performance, which would significantly improve computational efficiency relative to performance. As shown by comparing the second and third rows in the table below, even with a substantial reduction in channel size and computation, the performance improved.
>
> Comparing with traditional methods, the late split method in the first row shows competitive performance against models like Conv-TasNet, SuDoRM-RF, and DPRNN, achieving better performance with less computation using the Global-Local Transformer. The ESSD + CS (SepRe) structure in the third row demonstrated comparable or better performance with significantly reduced computation. Additionally, the proposed base model outperforms TF-GridNet with much smaller computation. This highlights the efficiency of the proposed SepRe method.
>
> We will make sure to clearly highlight these points in the revised paper to provide a better understanding of how the early split strategy impacts computational efficiency.
>
> | Case | F(scale) | Params. (M) | MACs (G/s) | SI-SNRi (dB) |
> | --- | --- | --- | --- | --- |
> | Ours + Late Split + origin dec | 64(Tiny) | 3.3 | 5.1 | 19.0 |
> | Ours + Late Split + origin dec | 128(Base) | 11.6 | 18.3 | 21.6 |
> | Ours + Early Split + shared dec +CS | 64(Tiny) | 3.7 | 10.4 | 22.4 |
> | Ours + Early Split + shared dec + CS | 128(Base) | 14.2 | 39.8 | 23.8 |
> | Conv-TasNet | - | 5.1 | 10.5 | 15.3 |
> | SuDoRM-RF | - | 6.4 | 10.1 | 18.9 |
> | DPRNN | - | 2.6 | 88.5 | 18.8 |
> | Sepformer | - | 26.0 | 86.9 | 20.4 |
> | SFSRNet | - | 59.0 | 466.2 | 22.0 |
> | ISCIT | - | 58.4 | 252.2 | 22.4 |
> | TF-GridNet | - | 14.5 | 460.8 | 23.5 |
>
> **Q4: Why was the alpha value set to 0.4, and have you experimented with other values to determine the optimal setting?**
>
> A4: We did not perform experiments with other values for the alpha parameter. While further tuning might potentially improve performance, we did not consider this aspect critical because the alpha value also decays as the training epoch proceeds. Therefore, we did not conduct extensive experiments to optimize this parameter.
>
> **Q5: How does the model handle varying lengths of input sequences, and what is the maximum sequence length it can effectively process?**
>
> A5: Thank you for this valuable question. During training, we fixed the input length to 4 seconds for efficiency. However, during evaluation, the model processes inputs of varying lengths in one batch, similar to most existing studies. We apologize for not explicitly stating this in the paper and will ensure to clarify this point in the revised version. The average length of test samples in the WSJ0-2Mix dataset is about 6 seconds, with a maximum length of about 15 seconds. The model can effectively separate sources within this range without significant issues. For sequences longer than 10-20 seconds, we can split the input into smaller segments of 5-10 seconds for processing.
>
> **Q6: Are there any limitations or failure modes of the proposed method that have been identified during experimentation?**
>
> A6: As mentioned in the conclusion, our method still struggles with cases involving varying numbers of speakers due to the fixed split layer. We believe this approach is important because, in practice, identifying the number of speakers in advance can be cumbersome. Our model still has limitations in this regard because it relies on a split layer with fixed input and output shapes. This means the network requires prior knowledge of the number of speakers to be separated, and individually trained models must have corresponding split layers to address different numbers of speakers. We will make sure to clarify these potential limitations in the revised paper.

---

> > ### Comment · Reviewer_aFMn · 2024-08-13
> >
> > Thank you for your responses. After considering your rebuttal, I maintain my score.

---

### Official Review · Reviewer_Gfro · 2024-07-13

**Soundness:** 3
**Presentation:** 2
**Contribution:** 3
**Rating:** 7
**Confidence:** 5

**Summary:**

The authors propose a neural network architecture to separate a speech mixture containing 2 speakers.  The proposed U-net based architecture replaces the inter and intra chunk processing - a popular method for speech separation - with global and local attention mechanisms. They further propose a mechanism to reduce the computational cost of the model. The model does the speech separation early in the network and uses a decoder with a low parameter count - all thanks to its weight sharing strategy - to do the speech separation.  Evaluation on simulated data shows improvement over the state of the art methods.

**Strengths:**

1. The Efficient Global Attention (EGA) component of the model, as discussed in Section 3.3, appears to be a cost-effective method for using the global context. This is achieved by initially subsampling to a reduced number of frames, thereby reducing the computational cost associated with transformers, and subsequently upsampling to a larger number of frames. The anticipated loss incurred from the downsampling process is compensated through the implementation of a gating mechanism.

2. Results show all the proposed mechanics, namely: early split, multi-loss, shared decoder parameters, EGA module design gives improvement in Si-SDR

3. Interestingly, when the proposed methods were applied to existing architectures such as conv-tasnet and Sepformer, an improvement in SI-SNR was observed.

4. Appendix D, also shows the computation effectiveness of the model

5. Attached samples clearly showed the quality of separated speech using the proposed network.

**Weaknesses:**

1. The authors have shown good results on a bunch of datasets, and the separated audios in the supplemental file are of high quality. But all these results are based on simulated data, so it makes you wonder how well the model would do with real data. It’d be great if the authors could show how the model performs on the Chime-6 dataset or Libricss, maybe through WER metrics or even objective speech perception measures after separation. Or at least, they could show us some samples after separation on real data.

**Questions:**

1. Looking at Figure 2, I’m wondering if it’s really a good idea to use the same split network for all ‘r’ values.  Would it be better to have a different ‘spk’ split for each ‘r’ ?

2. I believe Eq(1) should be a minimum of the Si-SNR and \tau since you refer to it in line 201?

3. In line 213, shouldn’t the multi loss be (1-\alpha / R) L + \alpha \sum_r L_r/R ?

4. It is not very clear  why the authors refer to the decoder as a Siamese decoder. Is this because they share the same weights across all speakers? This is typically how a speech separation network is structured. I don't see any contrastive learning loss functions as part of the training loss to discriminate the speakers.

**Limitations:**

Lack of evaluation on real data is a limitation of this work.

---

> ### Author Rebuttal · Authors · 2024-08-05
>
> **Q1: The authors have shown good results on a bunch of datasets, and the separated audios in the supplemental file are of high quality. But all these results are based on simulated data, so it makes you wonder how well the model would do with real data. It’d be great if the authors could show how the model performs on the Chime-6 dataset or Libricss, maybe through WER metrics or even objective speech perception measures after separation. Or at least, they could show us some samples after separation on real data.**
>
> A1: Thank you for your valuable feedback. We appreciate your point regarding the importance of evaluating the model on real-recorded datasets. We will include word error rate (WER) evaluation results for the LibriCSS dataset in the appendix.
>
> Additionally, we previously evaluated an early version of SepReformer on LibriCSS, similar to the proposed Ours+U-Net in Table 5. Although we cannot provide the WER from the latest SepReformer at this moment as it requires additional training with background diffuse noise and reverberation, we can share a comparison of an early version of SepReformer with DPRNN, using the Librispeech dataset and Room-Impulse-Response (RIR) simulations with background noise.
>
> | WER on LibriCSS |  |  |  |  |  |  |
> | --- | --- | --- | --- | --- | --- | --- |
> | Condition |  |  |  |  |  |  |
> | Overlap Ratio in %  | 0S | 0L | 10 | 20 | 30 | 40 |
> | Oracle | 4.9 | 5.1 | - | - | - | - |
> | Input | 11.8 | 11.7 | 18.8 | 27.2 | 35.6 | 43.3 |
> | DPRNN | 10.6 | 10.4 | 12.7 | 16.6 | 20.8 | 23.5 |
> | Early version of SepReformer | 9.8 | 10.1 | 10.9 | 12.5 | 14.4 | 17.5 |
>
> Please note that SepReformer is expected to perform better than the early version. Both DPRNN and our model were trained for stable separation and denoising without dereverberation.
>
> Furthermore, as our work focuses on speech, reporting metrics such as PESQ and STOI for datasets like WHAMR!, which include noise and reverberation, would be valuable. We will include PESQ and STOI results for the WHAMR! dataset in the appendix.
>
> | WHAMR! | SI-SDR (dB) | SDR (dB) | PESQ | eSTOI |
> | --- | --- | --- | --- | --- |
> | Unprocessed | -6.1 | -3.5 | 1.41 | 0.317 |
> | TF-GridNet[1] | 10.6 | 11.7 | 2.75 | 0.793 |
> | SepReformer-L | 11.0 | 12.5 | 2.77 | 0.796 |
>
> **Q2: Looking at Figure 2, I’m wondering if it’s really a good idea to use the same split network for all ‘r’ values. Would it be better to have a different ‘spk’ split for each ‘r’ ?**
>
> A2: Thanks for your insightful comment. In our early version, we initially used different split layers for each stage. We then considered using a shared split layer, assuming that consistently separated feature sequences could benefit the reconstruction decoder. Interestingly, we observed more stable convergence and no significant difference in separation performance. To simplify our model and reduce the number of parameters, we opted for the shared split layer. However, as you suggested, we will add experiments comparing the two approaches in the appendix, as exploring shared split layers for skip connections in the U-Net structure is indeed worth investigating.
>
> **Q3: I believe Eq(1) should be a minimum of the Si-SNR and \tau since you refer to it in line 201?**
>
> A3: We really appreciate for your detailed comment and apologize for the typo. As you commented, Eq(1) should be a minimum of the Si-SNR and \tau. We will correct it.
>
> **Q4: In line 213, shouldn’t the multi loss be (1-\alpha / R) L + \alpha \sum_r L_r/R ?**
>
> A4: If all the losses at each stage, including $\mathcal{L}$ should be considered equally, the multi-loss $\hat{\mathcal{L}}$ can be set to $(1-\alpha) \mathcal{L} /R + \alpha \sum_r \mathcal{L}_r/R$. However, we thought the final loss of $\mathcal{L}$ alone is important as much as summation of all the auxiliary loss $\mathcal{L}_r$. Therefore, we set the multi-loss as $(1-\alpha) \mathcal{L} + \alpha \sum_r \mathcal{L}_r/R$.
>
> **Q5: It is not very clear why the authors refer to the decoder as a Siamese decoder. Is this because they share the same weights across all speakers? This is typically how a speech separation network is structured. I don't see any contrastive learning loss functions as part of the training loss to discriminate the speakers.**
>
> A5: Thank you for your valuable comment, and we apologize for the confusion caused by misuse of the term. As you pointed out, we named the weight-sharing part across all speakers the Siamese decoder. We did not use an additional loss to discriminate the speakers using speaker identities. Instead, we trained the model directly with a separation objective based on  the Permutation Invariant Training (PIT) loss, which is conventional in speech separation network.
>
> Most speech separation networks are designed with a single feature sequence that is addressed before a late split layer. However, in our approach, given early separated feature sequences, whose similarity is relatively high (as indicated by grey line in Figure6(c)), the weight-sharing blocks can learn to discriminate speech sources (as indicated by orange and blue lines in Figure6(c)). This suggests that the discriminative learning is enhanced by the weight-sharing (or what we thought of as Siamese) structure. We believe that separation loss with PIT itself can operate as a kind of discriminative learning with ESSD structure, given the early split feature sequence.
>
> Nevertheless, we acknowledge that using the term ‘Siamese’ must be misleading since we did not apply a typical contrastive loss. Moreover, we concluded that prematurely using the term "Siamese" could be risky, as it may hinder considering scenarios involving more than two individuals in the future. Therefore, we will replace the term ‘Siamese’ with ‘weight-shared’ in the revised paper.
>
> [1] Z. -Q. Wang, S. Cornell, S. Choi, Y. Lee, B. -Y. Kim and S. Watanabe, "TF-GridNet: Integrating Full- and Sub-Band Modeling for Speech Separation," in IEEE/ACM TASLP, vol. 31, pp. 3221-3236, 2023

---

> > ### Comment · Reviewer_Gfro · 2024-08-13
> > **Thanks for addressing the comments.**
> >
> > I thank the authors for providing their rebuttal which address my concerns. I will retain the Accept score.

---

### Author Rebuttal · Authors · 2024-08-05

We would like to thank all the reviewers for their helpful comments and suggestions. We sincerely appreciate the time and effort they have dedicated to reading and commenting on the paper. Below, please find our point-by-point response to all the comments. We believe that it is greatly improved by incorporating the reviewers' suggestions and comments.

---

### Decision · Program_Chairs · 2024-09-25

**Decision:**

Accept (poster)

**Comment:**

The paper presented sepformer which uses global and local transformers with EGA and CLA to effectively model the global and local information for speaker separation. The proposed method has shown better results comparing to SoTA. The authors also provided additional results to address reviewers’ concerns. However, the presentation of the paper still needs improvements and the novelty is good but not breakthrough. Hence I recommend an accept.